# Cardiomyocyte-Specific Deletion of Orai1 Reveals Its Protective Role in Angiotensin-II-Induced Pathological Cardiac Remodeling

**DOI:** 10.3390/cells9051092

**Published:** 2020-04-28

**Authors:** Sebastian Segin, Michael Berlin, Christin Richter, Rebekka Medert, Veit Flockerzi, Paul Worley, Marc Freichel, Juan E. Camacho Londoño

**Affiliations:** 1Pharmakologisches Institut, Ruprecht-Karls-Universität Heidelberg, INF 366, 69120 Heidelberg, Germany; sebastian.segin@gmx.de (S.S.); michael.berlin@pharma.uni-heidelberg.de (M.B.); christin.richter@pharma.uni-heidelberg.de (C.R.); rebekka.medert@pharma.uni-heidelberg.de (R.M.); marc.freichel@pharma.uni-heidelberg.de (M.F.); 2DZHK (German Centre for Cardiovascular Research), Partner Site Heidelberg/Mannheim, 69120 Heidelberg, Germany; 3Experimentelle und Klinische Pharmakologie und Toxikologie, Universität des Saarlandes, 66421 Homburg, Germany; Veit.Flockerzi@uks.eu; 4The Solomon H. Snyder Department of Neuroscience, Johns Hopkins University, School of Medicine, Baltimore, MD 21205, USA; pworley@jhmi.edu

**Keywords:** calcium, cardiac remodeling, cardiac function, Orai1 proteins, neurohumoral stimulus

## Abstract

Pathological cardiac remodeling correlates with chronic neurohumoral stimulation and abnormal Ca^2+^ signaling in cardiomyocytes. Store-operated calcium entry (SOCE) has been described in adult and neonatal murine cardiomyocytes, and Orai1 proteins act as crucial ion-conducting constituents of this calcium entry pathway that can be engaged not only by passive Ca^2+^ store depletion but also by neurohumoral stimuli such as angiotensin-II. In this study, we, therefore, analyzed the consequences of Orai1 deletion for cardiomyocyte hypertrophy in neonatal and adult cardiomyocytes as well as for other features of pathological cardiac remodeling including cardiac contractile function in vivo. Cellular hypertrophy induced by angiotensin-II in embryonic cardiomyocytes from Orai1-deficient mice was blunted in comparison to cells from litter-matched control mice. Due to lethality of mice with ubiquitous Orai1 deficiency and to selectively analyze the role of Orai1 in adult cardiomyocytes, we generated a cardiomyocyte-specific and temporally inducible Orai1 knockout mouse line (Orai1^CM–KO^). Analysis of cardiac contractility by pressure-volume loops under basal conditions and of cardiac histology did not reveal differences between Orai1^CM–KO^ mice and controls. Moreover, deletion of Orai1 in cardiomyocytes in adult mice did not protect them from angiotensin-II-induced cardiac remodeling, but cardiomyocyte cross-sectional area and cardiac fibrosis were enhanced. These alterations in the absence of Orai1 go along with blunted angiotensin-II-induced upregulation of the expression of Myoz2 and a lack of rise in angiotensin-II-induced STIM1 and Orai3 expression. In contrast to embryonic cardiomyocytes, where Orai1 contributes to the development of cellular hypertrophy, the results obtained from deletion of Orai1 in the adult myocardium reveal a protective function of Orai1 against the development of angiotensin-II-induced cardiac remodeling, possibly involving signaling via Orai3/STIM1-calcineurin-NFAT related pathways.

## 1. Introduction

The main compensatory homeostatic responses to reduction of heart function are activation of the renin–angiotensin–aldosterone system (RAAS) and the sympathetic nervous system [1]. Angiotensin-II (AngII) is able to activate AT_1_− receptors which couple to G_q/11_, G_12/13_− or G_i/o_ and activates signaling cascades that involve the regulation of [Ca^2+^]_i_ and development of cardiac hypertrophy [2]. Besides its important role for contractile mechanisms, Ca^2+^ plays a fundamental function as a second messenger. It controls hypertrophic genes, e.g., via the calcineurin-NFAT-pathway by translocation of NFAT into the nucleus [3] and its homeostatic alterations underlie hallmarks (contractile dysfunction and arrhythmias) of heart failure [4]. Changes in intracellular Ca^2+^ concentration [Ca^2+^]_i_ and localization can be regulated by different pathways and triggers. Members of the TRPC subfamily of transient receptor potential canonical channels are involved in these processes (for review see [3,5,6,7,8]), and recently a background Ca^2+^ entry pathway (BGCE) was described in mouse adult cardiomyocytes, which is crucial for the pathological cardiac remodeling but does not modulate cardiac contractility in vivo. BGCE is significantly enhanced by AngII treatment and is mediated by TRPC1/TRPC4 channels [9]. Another crucial Ca^2+^ entry pathway into the cells, which can be evoked by receptor agonists leading to phospholipase C activation and depletion by IP_3_-sensitive Ca^2+^ stores, is the so-called store-operated Ca^2+^ entry (SOCE). This Ca^2+^ entry pathway was initially identified in non-excitable cells and it is mediated by Ca^2+^ channels that are activated by depletion of intercellular Ca^2+^ stores to refill Ca^2+^-depleted sarcoplasmic reticulum [10,11]. SOCE is driven by proteins of the Orai family. The Orai genes are widely expressed and encode transmembrane proteins that form highly selective Ca^2+^ channels and were first described in 2006 [12,13,14,15]. Three isoforms (Orai1, Orai2, and Orai3) are known [16,17] and Orai1 is a crucial component of SOCE in all cell types analyzed so far [18]. Ca^2+^ depletion is detected through STIM1 (Stromal Interaction Molecule 1) and STIM2 proteins, Ca^2+^ sensors located in the sarcoplasmic reticulum membrane [14,19]. STIM proteins bind Ca^2+^ as long as high sarcoplasmic reticulum Ca^2+^ concentrations are present. Following Ca^2+^ store depletion, the conformation of STIM proteins changes and the molecules oligomerize on the sarcoplasmic reticulum’s surface close to the plasma membrane to activate Orai proteins through direct contact and refilling Ca^2+^ storage begins [19,20,21,22].

SOCE has been described in neonatal and adult cardiomyocytes across different species including mice and rats, as well as in cardiomyocyte cell lines [23,24,25,26,27,28,29,30]. The amplitude of SOCE in adult cells was apparently lower in comparison to neonatal cardiomyocytes cells [23], but SOCE is re-activated in adult cardiomyocytes by mechanical and neurohumoral stress including AngII treatment; concordantly, cardiac expression of Orai1 and/or Stim1 is boosted in models of cardiac hypertrophy/remodeling, such as aortic banding [23], including transverse aortic constriction (TAC) in rats [24], TAC in mice [31,32], or related stress stimuli in neonatal rat ventricular cardiomyocytes [23,25]. SOCE and Stim1-dependent signaling in the heart and its implications in its physiology and pathophysiology has been matter of review [33,34,35,36].

Deletion of Orai1 proteins leads to cardiac dysfunction with reduced systolic function, bradycardia, and skeletal myopathy in zebrafish [31] and perinatal lethality in mice that can be rescued only partially by backcrossing to an outbred mouse strain [37]. Cardiac-specific suppression of Orai1 in Drosophila provoked reduced contractility consistent with dilated cardiomyopathy [38]. Heterozygous Orai1+/− mice of a C57BL6/J genetic background with ~25% reduction of Orai1 protein expression in the heart exhibit faster development of heart failure after TAC [39]. Initial evidence supporting the concept of a role of Orai1 in cardiac remodeling comes from the analysis of cellular models. It has been shown that Orai1 knockdown via siRNA in neonatal rat cardiomyocytes reduces SOCE and it is protective to hypertrophic stimuli such as phenylephrine [25]. Similarly, SOCE was also reduced in HL-1 cells by a similar approach [27]. More recently, the regulation of SOCE by aldosterone was proposed in neonatal rat ventricular myocytes where a dominant negative mutant of Orai1 or the Orai1 blocker S66 reduced aldosterone-induced SOCE [40]. Modulation of Orai1 through the NO (Nitric Oxide), cGMP (cyclic Guanosine Monophosphate), and protein kinase G (PKG) pathway, ending up in phosphorylation of Orai1 and reduction of SOCE, also abolished hypertrophic effects induced by phenylephrine in human embryonic stem cell-derived cardiomyocytes [41]. Orai1 expression was shown to be attenuated by gastrodin, resulting in reduced SOCE and protection from the hypertrophy-inducing agents’ phenylephrine and angiotensin-II, respectively [42]. In humans, Orai1 loss or gain of function mutations have been causally associated with myopathy and immunodeficiency [43].

To analyze the effect of complete inactivation of Orai1 for cardiac remodeling and function in the adult heart and to circumvent the potential effects of early deletion of Orai1 caused by global Orai1 deletion and mutation analysis in humans, we generated a cardiomyocyte-specific and temporally inducible Orai1 knockout mouse line. Analysis of cardiac contractility under basal conditions and after angiotensin-II infusion showed no differences between Orai1-deficient mice and control mice. Moreover, deletion of Orai1 selectively in cardiomyocytes did not evoke changes in cardiac histology and did not protect adult mice from AngII-induced cardiac remodeling. In contrast, AngII infusion resulted in increased cardiomyocyte cross-sectional area and collagen deposition, which were accompanied by differences in gene expression of Myoz2, STIM1, and Orai3 in the absence of Orai1. Thus, our results obtained from an approach with acute deletion of Ora1 in adult cardiomyocytes suggest that Orai1 proteins, unlike in embryonic cardiomyocytes, do not mediate processes of maladaptive cardiac remodeling in cardiomyocytes of adult mice in this type of cardiac pathophysiology but contribute to processes that are protective.

## 2. Materials and Methods

### 2.1. Animal Experiments

All animal experiments were approved and performed according to the regulations of the Regierungspräsidium Karlsruhe and the University of Heidelberg (AZ 35-9185.81/G131/15) and conform to the guidelines from Directive 2010/63/EU of the European Parliament on the protection of animals used for scientific purposes. Mice were maintained under specified pathogen-free conditions at the animal facility (IBF) of the Heidelberg Medical Faculty in a 12-h light-dark cycle, and water and standard food (Rod18, LASvendi GmbH, Soest, Germany) were available to consume ad libitum. Investigator and technical assistants were blinded towards genotype and treatment of the mice. Mice from single breeding pairs were randomly distributed in different treatment groups to avoid a breeding bias.

To generate Orai1+/− mice (used to obtain Orai1-/- embryos), we crossed Orai1^flox/flox^ mice [44] with a Cre-deleter strain [45]. To generate a cardiomyocyte-specific inducible Orai1 knock-out mouse model (Orai1^CM–KO^), we mated Orai1^flox/flox^ mice [44] with αMHC/CreERT2 mice kindly provided by Prof. Stefan Offermanns [46]. Male Orai1^flox/flox^/αMHC/Cre (either Cre^pos^ or Cre^neg^) mice were injected intraperitoneally with 1 mg of tamoxifen (T5648, Sigma-Aldrich Munich, Germany) dissolved in Miglyol (10 mg/mL) (Caelo, Hilden, Germany) for 5 consecutive days at an age of 11 weeks to induce deletion of exons 2 and 3 of the Orai1 gene. In part of the experiments, Miglyol at the same volume was given to control animals (Orai1^flox/flox^/αMHC/Cre^Pos^ mice used as additional control for possible Cre effects independent of tamoxifen), whereas in other experiments Orai1^flox/flox^/αMHC/Cre^neg^ mice treated with tamoxifen were used as control. Three weeks after the last injection, animals were analyzed under nontreatment conditions or after angiotensin II (Calbiochem/Merck, Darmstadt, Germany) treatment using osmotic Mini Pumps (Alzet model 1002, DURECT Corporation, Cupertino, CA, USA) at a rate of 3mg/kg/day as previously described [9]. Mini Pumps for the control group were filled with 0.9% NaCl.

To evaluate the αMHC/Cre efficiency, we used a αMHC-mTmG reporter mouse strain and determined the efficacy of Cre-mediated recombination and tissue specificity after injection of tamoxifen or Miglyol. To generate the reporter mice, αMHC/CreERT2 mice were crossed with mice from the mTmG (*Gt(ROSA)26Sor^tm4(ACTB-tdTomato,-EGFP)Luo^*/J) reporter strain [47]. Tamoxifen and Miglyol treatment were done, as mentioned above.

Pressure volume loop measurements were done under isoflurane anesthesia (2% with a mixture of O_2_/N_2_O (1:2), Vapor 19.3, Abbot). Pancuronium (1 µg/g body weight –BW-) and heparin (1.2 IE/g BW) were injected intraperitoneally. Mice were placed on a heating plate and temperature was monitored to keep it at 37 °C+/− 0.6 °C. During the complete procedure, mice were intubated and ventilated (MiniVent 845, Hugo Sachs Elektronik - Harvard Apparatus GmbH, March-Hugstetten, Germany) with tidal volume = 6.2 × body weight^1.01^, as described [48]. Adequate depth of anesthesia was proven by the absence of the paw withdrawal reflex from the hind paws before relaxation with pancuronium. For fluid maintenance and volume calibration, a catheter (outer diameter 0.61 mm) was inserted into the femoral vein and NaCl 0.9% was constantly perfused at a rate of 10 µL/min (11Plus, Hugo Sachs Elektronik - Harvard Apparatus GmbH, March-Hugstetten, Germany).

After thoracotomy a PVR 1035 catheter (Millar, Houston, TX, USA) connected to a MPVS Ultra (ADInstruments, Oxford, UK) was placed in the left ventricle after puncture with a 25-G needle. A suture around the inferior vena cava was used to change preload by transient occlusion of the vein for measuring the preload-independent parameters. While recording parameters with LabChart Pro 8 (ADInstruments, Oxford, UK), ventilation was stopped for a few seconds. At the end of the experiment, calibration was performed by injecting three single 10-µL boli of hypertonic saline (15%) and taking blood for volume cuvette calibration, as described [49]. For euthanasia, the ventilation was stopped, inducing a pneumothorax and the removal of the heart produced exsanguination, both occurred under anesthesia. Organs were analyzed for water content and weight, normalized to body weight and tibia length as described [9].

### 2.2. Cardiomyocyte Isolation

To obtain Orai1−/− embryonic cardiomyocytes, Orai1+/− mice were mated and pups were taken from plug-positive females at E18.5. Hearts were isolated separately, cleaned from noncardiac tissue residues in DPBS, Mg^2+^, and Ca^2+^ free (Thermo Fisher Scientifics/Gibco, Waltham, MA, USA), digested with trypsin 0.5% (Thermo Fisher Scientifics/Life Technologies) and DNase (2.2 kU/5 mL Trypsin, Sigma), and mechanically homogenized by repeated pipetting though a blunted tip. After centrifugation (Megafure 1.0, Heraeus, Hanau, Germany) at 319× *g* at 4 °C for 10 min, the cell pellet was resuspended in DMEM/F12 (Thermo Fisher Scientifics/Life Technologies) with 15% serum (Nu-Serum IV, BD Biosciences, San Jose, CA, USA) and seeded on cell culture plates (6-Well Cell Culture Plate, CellStar, Greiner bio-one; 10 cm^2^ per well, Greiner, Frickenhausen, Germany,). After 24 h, AngII (final concentration 100 nM in DPBS) was added to DMEM/F12 with 0.5% serum (FBS, 10270-106, Gibco) to induce cardiomyocyte hypertrophy. The control group received only DMEM/F12 with 0.5% serum. Cellular hypertrophy was compared in terms of change in cell area normalized to the mean value of the corresponding control condition.

Adult cardiomyocytes were isolated as previously described [9]. First, hearts were washed with oxygenated 200 µM EGTA containing perfusion buffer (134 mM NaCl, 11 mM Glucose, 4 mM KCl, 1.2 mM MgSO_4_, 1.2 mM Na_2_HPO_4_, 10 mM HEPES, pH 7.35) for 3 min. Then, hearts were perfused with oxygenated digestion buffer (perfusion buffer with 0.05 mg/mL Liberase TM, Roche Applied Science, Pleasanton, CA, USA) for 2–3 min. Atria were then separated from the ventricle and ventricular cells were dissociated by gentle pipetting. Tissue rests were removed by filtration through a 100-µm filter (Sysmex, Norderstedt, Germany) and extracellular Ca^2+^ concentration was stepwise increased to 1 mM. For expression analysis of enriched cardiomyocyte preparations, cells were purified by one decantation step (10 min/RT 22–24 °C) and two consecutive centrifugation steps (1 × 47 × *g* 1.5 min and 1 × 47 × *g* 1 min). Finally, PBS-washed pellets were resuspended in 500-µL TRIzol reagent (Thermo Fisher Scientifics/Invitrogen), frozen in dry ice, and stored at −80 °C until RNA isolation. For microscopy, cardiomyocytes were seeded on Extra Cellular Matrix-coated coverslips (ECM Gel from Engelbreth-Holm-Swarm murine sarcoma, E1270, Sigma). Afterwards, fixation was done with paraformaldehyde (PFA 4% in PBS pH 7.4, 10 min at 4 °C), until analysis cells were stored at 4 °C in PBS containing 0.2% Na-Azide.

Fluorescence images were obtained with an inverted microscope Z1 (Zeiss, Jena, Germany) equipped with a Digital Camera AxiocamMRm (Zeiss), a DG4 Plus/30 Lamp (Sutter Instrument Company, Novato, CA, USA), and appropriate DIC, 45-Texas Red and YFP filters (AHF analysentechnik AG, Tübingen, Germany), and control Software (AxioVision 4.8, multichannel module, Zeiss). For the analysis of adult cardiomyocytes from reporter mice (αMHC-mT/mG), images were taken with a Fluar objective (20×/0.75DICII (DIC.5-1.4), Zeiss) and the exposure times were 15–35 ms for bright field, 700 ms for red fluorescence, and 250 ms for the green fluorescence).

### 2.3. Analysis of Orai1 Expression and Cellular Hypertrophy by Immunocytochemistry

After fixation, cells were placed into −20 °C acetone for 5 min for permeabilization, then rinsed with chilled PBS and incubated at room temperature with 1% BSA (Carl Roth, GmbH, Karlsruhe, Germany) in PBST (PBS + 0.1% *v*/*v* Tween-20 and including 0.3 M Glycine). Cells were incubated over night with the primary antibody (anti-α-Actinin 1:800 (ACTN2) clone EA-53, Sigma) for cell size analysis and identification of cardiomyocytes or anti-Orai1 (#1003, 1:200) to detect Orai1 proteins [50] at 4 °C. Afterwards, cells were washed three times with chilled PBS. Incubation with the secondary antibody (1:200 either anti-rabbit AlexaFluor488 or anti-mouse AlexaFluor594, Thermo Fisher Scientifics/Invitrogen) was done for 2 h for anti-Orai1 and for 1 h for all other antibodies. Nuclei were stained using DAPI (4′,6-diamidino-2-phenylindole, Thermo Fisher Scientifics/Invitrogen) (1.5 µg/mL in PBS for 5 min). During incubation times, the cells were protected from light to avoid fading. Afterwards, cells were mounted and covered in anti-fade (6 g Glycerin, 2.4 g Mowiol 4–88, 6 mL ddH_2_O, 12 mL Tris-HCl 0.2 M pH 8.5, and DABCO 25 mg/mL solution). Cell size was analyzed after staining using the α-Actinin antibody under a fluorescence microscope (Z1, Zeiss) using the quantification tool of the AxioVision software 4.8 (Zeiss).

### 2.4. RNA Isolation and qPCR Analysis

RNA isolation from Langendorff heart-derived cell populations or heart samples was performed using the TRIzol reagent (Thermo Fisher Scientifics/Invitrogen) including two final ethanol washing steps before RNA resuspension. When required, isolated RNA was further purified by 3–5 additional extraction steps with one volume of diethyl ether and a final ethanol precipitation with 0.3 M potassium acetate. DNA digestion was performed with the Turbo DNA-free^TM^ kit (Thermo Fisher Scientifics/ambion) following the instructions of the manufacturer with two DNAse incubation steps. Purity, integrity, and quantity of the RNA samples were controlled by spectrophotometry (NanoQuant, Tecan, Männedorf, Switzerland), microfluidic analysis (BioAnalyzer 2100, Agilent Technologies, Santa Clara, CA, USA), and fluorometry (qbit assay, ThermoFisher Scientific), respectively. RNA was separated in several aliquots and stored at −80 °C in order to use only freshly thawed RNA for qPCR experiments. Then, cDNA synthesis from 1000 ng of RNA was carried out using a SuperScript first-strand synthesis system for RT-PCR (SensiFAST BIO-65054, BIOLINE, London, UK). Fifty ng cDNA from each sample were used as template. In brief, qPCR was performed with a Universal Probe Library (Roche) (probes and primers listed in Appendix A) by using Roche FastStart Essential DNA Probes MasterMix (Roche 06402682001) and detection on a Light Cycler96 (Roche). Thermal cycling conditions included: Initial denaturing at 95 °C for 600 s, followed by 40 cycles of amplification (each cycle: 10 s/95 °C, 30 s/60 °C). The standard curve quantization method was used for gene expression measurements. Values are expressed as relative expression and calculated as follows:Relative Expression = (Mean E^−Cq^/Mean E_HKG_^−Cq^)/((Mean E_HKG_/100+1)^-Cq^×10^4^)(1)
with E being the efficiency, Cq the cycle of quantification, and HKG refers to housekeeping genes.

All PCR reactions were performed in duplicate and normalized to the expression of the housekeeping genes H3F3A, AIP, and CXXC1, as done before [51]. In case of heart samples, only the H3F3A and CXXC1 genes were used for normalization since AngII treatment produced a significant upregulation of the AIP gene in control mice.

### 2.5. Histological Analysis

Briefly, and as described before [9], cardiac tissue was fixed for 24 h with 4% paraformaldehyde in PBS (pH 7.4). Processing of the tissue was done in a Tissue Processor (Leica TP 1020, Leica Biosystems, Wetzlar, Germany) over 24 h and samples were manually embedded in paraffin. For cross-sectional area analysis and cardiac fibrosis, heart sections were stained with hematoxylin eosin (HE) or Sirius red (SR), respectively. Images were taken and analyzed with AxioVision 4.8 (Zeiss).

### 2.6. Statistical Analysis

Data were analyzed with Excel 2011 (Microsoft, Redmond, WA, USA) and Origin 8 (OriginLab Corporation, Northampton, MA, USA). Values are shown as mean ± standard deviation. Statistical significances were assumed when *p* < 0.05. The *p* values are depicted as * *p* < 0.05, ** *p* < 0.01, and *** *p* < 0.001. Statistical analysis was performed using the appropriate tests (unpaired Student’s t-test, two-way ANOVA, Kruskal–Wallis) after normality test (Shapiro–Wilk normality test). Two-way ANOVA tests were followed by Bonferroni’s post hoc comparisons to test specific pair comparisons and Dunn’s test after Kruskal–Wallis.

## 3. Results

### 3.1. Orai1 Deletion Protects from Neurohumorally Induced Cellular Hypertrophy in Embryonic Cardiomyocytes

Since global Orai1 deletion leads to perinatal lethality in mice [37,52], we utilized prenatal cardiomyocytes (E18.5) for the evaluation of the role of Orai1 in the development of cellular hypertrophy in vitro. We intercrossed Orai1+/− mice to obtain Orai1−/−, Orai1+/−, and Orai1+/+ cardiomyocytes. After immunocytochemical analysis of cells stained with anti-Orai1 antibody, we detected expression of Orai1 proteins in cells from wild-type (WT) mice. Specificity of the immunostaining was demonstrated in cells from Orai1−/− embryos. Orai1 expression was also detected in Orai1+/− cells (Figure 1A). Here, cardiomyocytes were identifiable through staining for α-Actinin, confirming the expression of Orai1 in cardiomyocytes and in α-Actinin-negative cardiac cells at this developmental stage. In cells isolated from Orai1−/− embryos, the hypertrophic stimulation using angiotensin-II (AngII) did not result in an increase of cardiomyocyte area, in contrast to Orai1+/+ and Orai1+/− cells, which both showed a significant rise in cell size (Figure 1B). In our hands, angiotensin-II alone was effective in inducing hypertrophic effects in cardiomyocytes in WT cells and produced a stronger effect (*p* < 0.001) than phenylephrine after 72 h (Appendix A).

### 3.2. Unaltered Cardiac Function in Cardiomyocyte-Specific Orai1-Deficient Mice

Firstly, we analyzed the efficiency of the Cre recombinase expressed under the cardiomyocyte specific promoter αMHC assessing Cre-dependent GFP fluorescence in isolated adult cardiac myocytes and we found GFP expression only in mTmG/αMHC-Cre^pos^ mice (Appendix A) with 99.7% of the cardiomyocytes being GFP-positive; notably, no GFP-positive cells were observed in the control group. At the tissue level Cre activity was observed only in cardiac tissue and in a restricted portion of large-diameter blood vessels of the lungs (Appendix A). All together, these results indicate a highly efficient and restricted expression pattern of the αMHC-Cre transgene used here for Orai1 deletion in cardiomyocytes. To further analyze the role of Orai1 in adult cardiomyocytes, we only used a tamoxifen-Cre-based inducible model for specific deletion of Orai1 in adult mice (referred always as Orai1^CM-KO^). With this model, we pursued to circumvent the lethality during early developmental stages or possible developmental effects if Orai1 is constitutively deleted in the heart in early stages. In isolated adult cardiomyocytes from Orai1^CM–KO^ (Orai1^flox/flox^/αMHC/Cre^pos^) mice we observed a reduction by 81.4 ± 5.4% in Orai1 transcript levels compared to controls (Orai1^flox/flox^/αMHC/Cre^neg^) (Figure 2A). We performed immunocytochemistry stainings in cardiomyocytes from adult WT and Orai1^CM–KO^ mice using different anti-Orai1 antibodies. However, we were not able to specifically observe its expression in these adult cells by this method (data not shown). The remaining Orai1 transcripts were likely to stem from noncardiomyocyte cells, as it is known that one cannot entirely get rid of nonmyocytes by this isolation procedure. In addition, we analyzed the expression of other genes important in intracellular Ca^2+^ signaling, and observed in Orai1^CM-KO^ cardiomyocytes a significant downregulation of Orai3 (by 17 ± 4.3%), Stim2 (by 30.2 ± 6.6%), and Saraf (by 16.5 ± 6.5%) expression (Figure 2A). In contrast, no significant changes were observed in the expression of Orai2, Stim1, Trpc1, Trpc3, Trpc4, and Trpc6 (Figure 2A). We then analyzed whether Orai1 deletion in cardiomyocytes provoked any allometric change. The allometric analysis (Appendix A) showed that Orai1^CM–KO^ mice were lighter (control 30.22 g vs. Orai1^CM–KO^ 27.72 g, *p* = 0.046) and showed lighter livers (1461 mg vs. 1251 mg, *p* = 0.011) and increased liver water content (66.8% vs. 67.9%, *p* = 0.015). The heart weight was comparable between both groups (133 mg vs. 129 mg, *p* = 0.481), also in relation to the tibia length (Figure 2B). Here, the reduction of the heart weight to body weight ratio is explained by the reduction in body weight of Orai1^CM–KO^ mice. Analysis of cardiomyocyte cross-sectional area and cardiac fibrosis were not different between both genotypes (Figure 2C). In addition, all parameters of cardiac contractile function including cardiac output, preload recruitable stroke work (PRSW), and ejection fraction, as well as volumetric parameters, such as end-diastolic pressure volume relationship (EDPVR), were unaltered in Orai1^CM–KO^ mice (Figure 2D) compared to control animals. Also, other systolic and diastolic (e.g., Tau) parameters obtained from PV loop measurements remained unaffected by Orai1 deletion in cardiomyocytes (Table 1).

### 3.3. Deletion of Orai1 in Cardiomyocytes Aggravates the Outcome of Morphometric and Functional Parameters after Angiotensin-II-Induced Cardiac Remodeling

After two weeks of systemic administration of AngII to Orai1^CM-KO^ mice an allometric analysis was performed (Appendix A). A significant increase in heart weight (136.31 mg in control/NaCl vs. 173.7 mg in control/AngII *p* < 0.001, 140.69 mg in Orai1^CM-KO^/NaCl vs. 186.66 mg in Orai1^CM-KO^/AngII, *p* < 0.001) accompanied by significant reduction in body weight was observed in both genotypes after AngII treatment. The hypertrophy indices heart weight/body weight and heart weight/tibia length were similarly increased by AngII in both genotypes (Figure 3A). In contrast, histological analysis of cardiac sections revealed that chronic AngII treatment leads to an increase in cardiomyocyte cross-sectional area and in fibrosis (2-way ANOVA, *p* < 0.0001), and that AngII-evoked cardiomyocyte hypertrophy (Figure 3B) and fibrosis (Figure 3C) were aggravated upon Orai1 deletion in cardiomyocytes. Beyond that, liver weight was reduced significantly by the AngII treatment (2-way ANOVA, *p* = 0.0003) independent of the genotype (*p* = 0.0597, 1572.03 mg in control/NaCl vs. 1303.15mg in control/AngII *p* < 0.01, 1406.77 mg in Orai1^CM-KO^/NaCl vs. 1262.26 mg in Orai1^CM-KO^/AngII, *p* = 0.308), even though the water content was increased (66.35% vs. 69.09% p < 0.001, 66.80% vs. 68.94% *p* < 0.001). Expression analysis of hypertrophy markers showed that βMHC mRNA levels were significantly higher in hearts from Orai1^CM–KO^ mice at the end of AngII treatment compared to control mice (Figure 3D). Other hypertrophy markers such as ANP, BNP, and the skeletal-α-actin were similarly increased in both genotypes (Figure 3D). The fibrosis markers including TGF-β2, TGF-β3, CTGF, Collagen -I and Collagen-III were similarly increased after AngII in both genotypes (Figure 3E).

Analysis of parameters of cardiac contractility revealed that AngII treatment did not evoke changes in EF, cardiac output, and PRSW in the control group (Figure 4B,C). However, the ejection fraction in Orai1^CM-KO^ mice was reduced independently of the treatment (2-way ANOVA, *p* = 0.0010), but cardiac output, PRSW, and EDPVR were not significantly altered between genotypes. Remarkably, after AngII treatment, the end-diastolic volume was decreased in Orai1^CM-KO^ mice (21.85 µL vs. 16.11 µL *p* = 0.025) but not in controls (18.15 µL vs. 15.63 µL *p* = 0.319) as well as the stroke volume (16.16 µL in Orai1^CM-KO^/NaCl vs. 11.41 µL in Orai1^CM-KO^/AngII *p* = 0.025), which also remained unchanged upon AngII treatment in control mice (13.46 µL vs. 13.57µL, *p* = 0.631) (Table 2). Finally, diastolic parameters (Tau, EDPVR) were unaffected by both AngII treatment and Orai1 deletion (Figure 4C and Table 2).

### 3.4. Differential Gene Expression in Orai1^CM-KO^ Hearts after Neurohumorally Induced Cardiac Remodelling

Finally, we analyzed if Orai1 deletion in cardiomyocytes affects the expression of (1) genes encoding proteins with known impact on SOCE activity, (2) genes encoding TRPC channels that might functionally interact with Orai1 channels [53], and (3) genes that may be activated downstream of Orai1 as Ca^2+^-dependent signaling molecules determining pathological cardiac remodeling. AngII treatment produced a significantly increased expression of Orai1 and Stim1 in control but not in Orai1^CM-KO^ hearts (Figure 5A), indicating a relatively restricted regulation of these genes to cardiomyocytes under AngII treatment and control of Stim1 expression by Orai1. In contrast, Orai2 expression was similarly increased by AngII in both groups, whereas Stim2 and Saraf expression were unaffected by AngII treatment (Figure 5A). Like Stim1, Trpc4, and Trpc6 expression increased in control mice treated with AngII but not in Orai1^CM-KO^ mice. Expression of Trpc1 and Trpc3 was unaffected by both genotype and AngII treatment (Figure 5B). Remarkably, we observed that the increased expression of the negative regulator of calcineurin, Myoz2 (Calsarcin-1), in AngII-treated control mice was completely absent in Orai1^CM-KO^ mice; additionally, the effect of AngII observed in controls hearts on Rcan1, another calcineurin regulator, was significantly reduced in Orai1^CM-KO^ hearts without changes in the calcineurin A alpha expression (Ppp3ca) (Figure 5C). In contrast, similar effects between control and Orai1^CM-KO^ mice were observed after AngII-induced cardiac remodeling in regards to Mef2a downregulation and increased expression of one of its targets, Myomaxin, (Figure 5C), suggesting no obvious involvement of Orai1 on MEF2a-dependent transcriptional activity.

## 4. Discussion

During heart failure the initially compensatory activation of the renin-angiotensin-aldosterone (RAAS) system turns into a chronic deleterious stimulus for the cardiovascular system; this chronic neurohumoral stimulation includes different important remodeling mechanisms, which are, in part, studied in models of chronic AngII treatment [1,54]. Therefore, we analyzed here the role of Orai1 under AngII stress. Knockdown of Orai1 has been reported to partially reduce the development of neurohumoral-induced cellular hypertrophy in cultured rat neonatal cells [25]. In this study, we aimed to assess whether Orai1 indeed contributes to cardiomyocyte hypertrophy and other features of pathological remodeling in vivo. We showed that the genetic deletion of the Orai1 gene is sufficient to completely abolish AngII-induced hypertrophy in mouse embryonic cardiomyocytes. To circumvent potential effects of Orai1 deletion during early development, we generated and studied a cardiomyocyte-specific and temporally inducible Orai1 knockout mouse line. Induction of Orai1 deletion in adult mice did not evoke changes in cardiac contractility or allometric parameters. Counterintuitively, Orai1 deletion in cardiomyocytes aggravated AngII-induced cardiomyocyte hypertrophy and fibrosis.

### 4.1. In Vitro Analysis of Orai1 in Embryonic Cardiomyocytes

We reported here that complete deletion of Orai1 in embryonic mouse cardiomyocytes (E18.5) blunted the AngII-induced hypertrophy. In addition, we showed that at this embryonic stage the cardiomyocytes express Orai1 proteins and we included cells from Orai1-deficient mice in an analysis to investigate the specificity of our anti-Orai1 antibody. Importantly, cells isolated from heterozygous embryos showed detectable Orai1 expression and developed similar cellular hypertrophy in response to AngII, showing that one allele is sufficient to fully mediate Orai1-mediated cellular hypertrophy. Nevertheless, in this cellular model, critical parameters with impact on remodeling in the whole heart in vivo-like wall stress, synchronization of electrical activity, the influence of systemic catecholamines, or the contribution of nonmyocytes (e.g., fibroblasts, macrophages, or immune cells) were not taken into account. Therefore, we used a Cre/loxP-mediated approach to delete Orai1 in adult cardiomyocytes since a systemic pharmacological approach that allows selective and efficient inhibition of Orai1 channels in cardiomyocytes is not feasible.

### 4.2. Impact of Specific Cardiomyocyte Deletion of Orai1 on the Adult Heart

To causally determine the role of Orai1 in the adult heart we inactivated Orai1 expression in adult cardiomyocytes. We included different control groups to determine the effect of tamoxifen or Cre expression (please see short discussion in Appendix A). Based on our experiments with mTmG reporter mice [47], we expected to have an efficient Cre-mediated recombination in adult cardiomyocytes in more than 99% of the cells. We observed by qPCR a reduction in *Orai1* expression by 52 ± 13% in heart samples and 81.4 ± 5.4% in cardiomyocytes isolated by Langendorff perfusion from Orai1^CM-KO^ mice. The remaining expression most likely reflected the contribution of nonmyocytes as Orai1 expression in α-Actinin-negative cells isolated from embryonic heart was also found in our study.

The analysis of cardiac contractility under basal conditions by PV loop analysis in Orai1^CM-KO^ mice in comparison to the control group (Orai1^flox/flox^/αMHC-Cre^pos^ mice treated with Miglyol) was performed 4 weeks following induction of Orai1 deletion, when transient effects of tamoxifen treatment on cardiac function [55] were not expected. No differences in cardiac function were observed in Orai1^CM-KO^ mice. Specifically, cardiac output, ejection fraction, EDPVR, or preload recruitable stroke work were unaltered. Histological analysis of cardiomyocyte cross-sectional area and collagen deposition (fibrosis) were also unchanged. No signs of heart failure with reduced ejection fraction or of heart failure with preserved ejection fraction (diastolic dysfunction) could be detected in Orai1^CM-KO^ mice. The absence of systolic or diastolic dysfunction in Orai1^CM-KO^ mice was in contrast with the effect of the deletion of Stim1, a critical regulator of Orai1 activity that leads to reduced cardiac function under basal conditions already 40 days after induction of Stim1 deletion [56]. These differences in the phenotype between Orai1^CM-KO^ mice and Stim1^CM-KO^ mice indicate that STIM1 has targets additional to Orai1 in cardiomyocytes, which might include voltage-gated Ca^2+^ channels [57,58]. Nonetheless, we cannot discard long-term effects of Orai1 deletion in the adult heart during aging, like those observed in other mouse model with cardiomyocyte-restricted ablation of Stim1 [59,60].

A possible reason for the absence of obvious changes in cardiac function in Orai1^CM-KO^ mice might be that the function of Orai1 is reduced in naïve adult cardiomyocytes [33]; reduction in expression of Orai1 in the adulthood was already reported in mice [31]. Another possibility is that the lack of Orai1 was compensated by the other members of the Orai family, such as Orai2 or Orai3. Whereas there are no reports about the functional role of Orai2 in cardiomyocytes or for cardiac function, Orai3 regulates a constitutive Ca^2+^ entry in adult left ventricular cardiomyocytes [33,61]. In an in vivo model where Orai1 (also Orai3) expression was reduced by and siRNA-based approach in adult ventricular rat cardiomyocytes, the expression of Orai3 was increased under basal conditions and after abdominal aborting banding; moreover, the constitutive Ca^2+^ entry in adult left ventricular cardiomyocytes was increased after Orai1 silencing and abolished when Orai3 was silenced [61]. Very recently it was briefly reported that cardiomyocyte-specific deletion of Orai3 causes dilated cardiomyopathy in mice at 4 months of age and the cardiac remodeling was more aggravated in younger mice during aortic banding [62]. In isolated cardiomyocytes from Orai1^CM-KO^ mice, we observed unchanged expression of *Orai2* compared to controls and a downregulation of *Orai3* (together with *Stim2* and the SOCE-associated regulatory factor, *Saraf*), pointing out a tight and mutual regulation in expression of these constituents and regulators of SOCE. On the other hand, the amplitude of store-operated Ca^2+^ entry (SOCE), which correlates with the abundance of Orai1 [25], was significantly lower in adult compared to embryonic cardiomyocytes [23] and, accordingly, the lack of Orai1 may be dispensable for cardiac function in adult mice. Since SOCE was reactivated following induction of hypertrophy by neurohumoral stimuli, including AngII treatment in adult cardiomyocytes [23], we next challenged Orai1^CM-KO^ mice with chronic AngII treatment.

We and others observed changes in the expression of Orai subtypes and of other regulators of SOCE when the expression of one of those proteins was altered; in addition, there was a complex mutual regulation by individual Orai subtypes as observed in cells from Orai1- and Orai2-single KO mice and Orai1/2 double-deficient mice as shown in different T cells subtypes or in mast cells [63,64]. Thus, to obtain a deeper understanding of the contribution of Orai proteins in cardiac remodeling, more expansive approaches will be required that include single and compound KO models with cardiomyocyte-specific deletion of genes encoding the corresponding Orai subtypes.

### 4.3. Protective Function of Orai1 Proteins in Adult Cardiomyocytes During Neurohumorally Induced Cardiac Hypertrophy

In contrast to the expectation from embryonic cardiomyocytes, where hypertrophy was blunted when Orai1 expression was entirely abrogated by Orai1 deletion, cardiac hypertrophy indices were not different in adult Orai1^CM-KO^ mice compared to control mice, and, remarkably, the interstitial collagen deposition as well as the cardiomyocyte cross-sectional area were significantly increased. From the comparison of absolute values of some functional parameters (i.e., EF) between Figure 2 and Figure 3, some discrepancies might be presumed, but it is important to note that the animals analyzed in these experimental groups differed in the fact that we used Orai1^flox/flox^/αMHC-Cre^Pos^ (+ Miglyol) as control mice in the first one (Figure 2) and Orai1^flox/flox^/αMHC-Cre^Neg^ (+ tamoxifen) were used as controls in the latter (Figure 3). The reason for these different approaches was that we were concerned and aimed to check for possible tamoxifen effects in our remodeling model. These differences could indicate, in general, that beyond the effect of AngII there could be also an effect of Cre (independent of tamoxifen) and point to the importance of different types of controls during this kind of experiment.

We observed a significant reduction in the ejection fraction in Orai1^CM-KO^ mice at the end of the two weeks of chronic AngII infusion. Accordingly, stroke volumes were also decreased in Orai1^CM-KO^. These results suggest a rather protective function of Orai1 in adult cardiomyocytes for the maintenance of proper cardiac function under stress conditions. Recently, Horton et al. reported that targeting of one Orai1 allele by a gene-trap approach, which goes along with partial reduction of Orai1 expression in the heart, led to escalation of heart failure development and an increase in mortality evoked by chronic pressure overload [39]. In this study the influence of Orai1 in cardiac non-myocytes or extracardial cells may also contribute to the observed phenotype and it remains to be analyzed in mouse lines in which Orai1 deleted other cell types relevant for cardiac remodeling, like cardiac fibroblasts or macrophages. Additionally, there were discrepancies between the gene-trapping approaches that most likely generated a hypomorphic Orai1 allele [65] compared to the mouse model generated by defined deletion of exons 2 and 3 by homologous recombination [37]. Very recently it was shown that Orai1 inhibition in adult murine cardiomyocytes by the expression of a dominant-negative mutant (Orai1^R91W^) or by pharmacological treatment ameliorated the reduction in systolic function provoked by pressure overload [66]. This approach differed from ours as expression of a dominant negative Orai1 subtype may affect all channel entities consisting of Orai1 proteins, and the composition of the channel complex triggered by AngII in cardiomyocytes is still unknown. In addition, the phenotype obtained by systemic pharmacological inhibition of Orai1-containing channel complexes can also be mediated by cardiac nonmyocytes and extracardiac actions of this substance. Moreover, the signaling pathways contributing to AngII-induced remodeling as analyzed in our study may differ compared to the pressure overload-evoked remodeling process as seen in AngII Type 1A receptor KO mice, which are protected from AngII-induced hypertrophy but not from pressure overload hypertrophy [67]. Despite the finding from this former study that Orai1 is a critical mediator for Ca^2+^ entry and cardiac remodeling, the composition of the channel complex triggered by AngII in cardiomyocytes is still unknown. Nevertheless, both the study by Bartoli et al. and ours are in agreement with the concept of Orai1 as a critical constituent of the Ca^2+^ entry pathway mediating cardiomyocyte and cardiac remodeling with Orai1 acting in concert with other channel proteins that can take over and even overcompensate when Orai1 is absent. Based on our results obtained by selective deletion of Orai1, it is well conceivable that other constituents of the channel complex, such as Orai2 and/or Orai3, might take over. Orai1 is known to interact with Orai2 or Orai3 in several cell systems functionally as well as physically, and such behavior has been reported by others and by us in lymphocytes and mast cells, respectively [63,64].

### 4.4. Cardiac Transcriptional Changes in the Absence of Orai1 in Cardiomyocytes after AngII-Evoked Cardiac Remodelling

Transcriptional upregulation of Orai1-3 and reactivation of SOCE amplitude under stress condition was reported previously, including upregulation of SOCE activity after TAC [23] and upregulation of Orai1 mRNA and protein expression after TAC and myocardial infarction [31]. These results imply that Orai-mediated signaling is part of the compensatory signaling cascade that includes control of transcriptional programs and complex Ca^2+^ homeostatic regulatory mechanisms during the development of pathological cardiac hypertrophy in adulthood [68]. We observed that the upregulation of *Orai3* and *Stim1* expression evoked by neurohumoral stress via chronic AngII treatment depends completely on the expression of Orai1. Previous work in immune cells [69] but also in neonatal cardiomyocytes [31] established that the signaling cascade downstream of Orai1 is associated with the activation of the protein phosphatase calcineurin. Calcineurin is a key player in the development of cardiac hypertrophy [70] and it is able to activate two important transcriptional regulatory pathways, one through the nuclear factor of activated T cells (NFAT) and the other through the myocyte enhancer factor 2 (MEF2) [71,72].

Orai1-deletion in cardiomyocytes does not seem to affect MEF2a signaling as expression of *Mef2a* itself and its target *Myomaxin* was similarly regulated in control and Orai1^CM-KO^. An increase in AngII-evoked upregulation of Myomaxin was found to depend on TRPC1/TRPC4 that mediates a Ca^2+^ entry in cardiomyocytes described as BGCE (background Ca^2+^ entry pathway). TRPC1/TRPC4 deletion together with reduced BGCE preclude AngII-evoked rise in electrically triggered Ca^2+^ transients and decreases AngII-induced hypertrophy and fibrosis in vivo [9].

On the other hand, our expression analysis in hearts of Orai1^CM-KO^ mice suggests that Orai1 may contribute to the calcineurin-NFAT signaling axis as AngII-evoked RCAN1 upregulation is reduced in Orai1^CM-KO^ hearts. The current study showed a reduced expression of *Rcan1* and enhanced cardiac remodeling in the absence of Orai, whereas reduced *Rcan1* expression in the absence of TRPC1/TRPC4 led to a decrease in the development of cardiac hypertrophy [9]. Based on these observations, it is tempting to speculate how two distinct Ca^2+^ entry pathways in cardiomyocytes could regulate the development of the cardiac remodeling differentially. The results contribute to understand the proposed involvement of transcriptional pathways regulated by the calcineurin-NFAT and/or the CaMKII (Ca^2+^/calmodulin-dependent protein kinase II)-MEF2 axis. This is not surprising since RCAN1 has been described to play a dual role during the development of cardiac hypertrophy as it can either facilitate or suppress cardiac calcineurin signaling [73].

Additional and yet to be discovered are properties of Orai1 and TRPC1/TRPC4 proteins that are not directly related to their Ca^2+^-conducting function. It is still not clear to which extent the solely Ca^2+^-conducted-by-cation channel is responsible for the transcriptional regulation and functional/morphological changes observed during cardiac remodeling. It is plausible that downstream signaling, including transcriptional changes, is regulated independently of the channel function, as identified in other Ca^2+^ channels [74,75]. Nevertheless, the upregulation of the negative regulator of calcineurin, *Myoz2* (*Calsarcin-1*) was abrogated in the absence of Orai1 in AngII-treated hearts. This might give an additional hint how cardiomyocyte hypertrophy and fibrosis may come about in Orai1^CM-KO^ mice after AngII infusion as calsarcin-1-deficient mice exhibit stress-evoked cardiomyocyte hypertrophy as a consequence of inappropriate calcineurin activation [76,77].

Together, our results reveal a critical role of Orai1 for fine-tuning cardiac remodeling. They indicate an inverse mode of action between early developmental stages and disease progression in the adulthood, as it operates as a crucial mediator for the hypertrophic response in embryonic myocytes but as a limiting determinant in hypertrophy and fibrosis in neurohumorally evoked cardiac pathology in adults.

## Figures and Tables

**Figure 1 cells-09-01092-f001:**
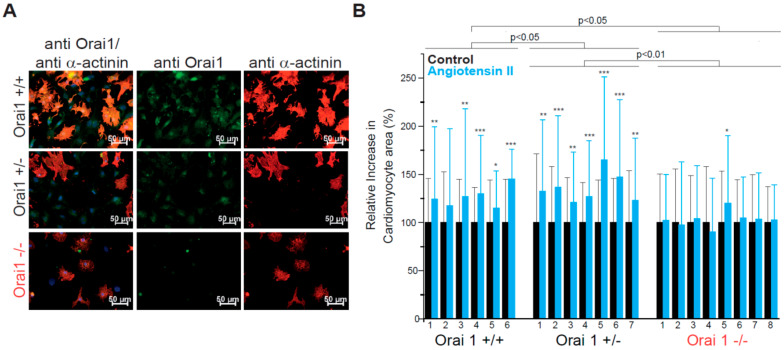
Orai1 is expressed in embryonic murine cardiomyocytes and its genetic deletion protects from neurohumoral-induced cellular hypertrophy. (**A**) Detection of Orai1 proteins (green) by immunocytochemistry in embryonic cardiomyocytes. As control for antibody specificity, cardiomyocytes from global Orai1-deficient (Orai1−/−) mice were used. Staining with α-Actinin (red) was used to identify cardiomyocytes and DAPI (blue) to stain the nuclei. (**B**) Relative increase of cardiomyocyte area after angiotensin-II stimulation (100 nM, 72 h). Orai1+/+ (n = 6 hearts) and Orai1+/− (n = 7 hearts) cardiomyocytes showed a significant increase in cell size while Orai1−/− (n = 8 hearts) cells remained at a similar size. Number under bars indicates individual animals; **p* < 0.05, **p < 0.01, ****p* < 0.001 for comparison between control and treatment from each isolation/heart and depicted p-values comparing between genotypes were obtained after Kruskal–Wallis and Dunn’s comparisons.

**Figure 2 cells-09-01092-f002:**
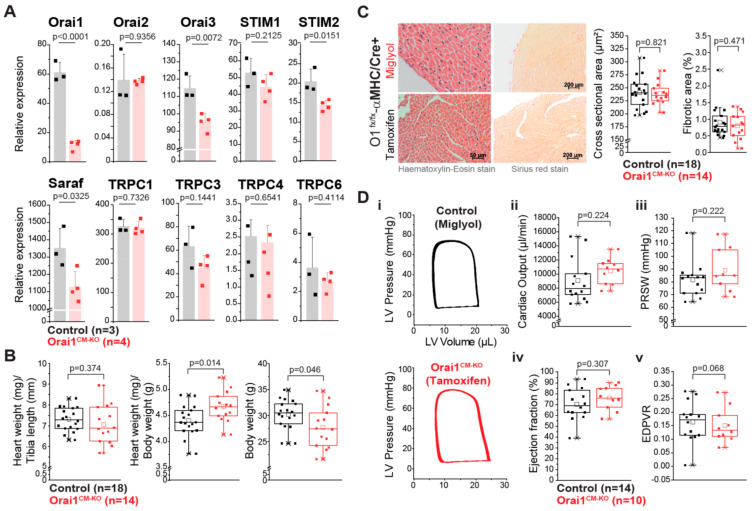
Mice with cardiomyocyte-specific deletion of Orai1 exhibit normal cardiac function. (**A**) Expression analysis of Orai1, Orai2, Orai3, STIM1, STIM2, SARAF, and TRPCs transcripts from enriched adult cardiomyocyte preparations of Orai1^CM-KO^ mice (Orai1^flox/flox^/αMHC-Cre^pos^ + tamoxifen) by qPCR and compared to preparations from control mice (Orai1^flox/flox^/αMHC-Cre^pos^ + Miglyol). (**B**) Analysis of basal cardiac hypertrophy indexes and body weight in Orai1^flox/flox^/αMHC-Cre^pos^ mice receiving Miglyol (Control) or tamoxifen (Orai1^CM–KO^). (**C**) Analysis of cardiomyocyte cross-sectional area and ventricular fibrosis from mice shown in (**B**). (**D**) In vivo cardiac function in Orai1-deficient mice (Orai1^flox/flox^/αMHC-Cre^pos^ + tamoxifen) analyzed by PV loop analysis and compared to control mice (Orai1^flox/flox^/αMHC-Cre^pos^ + Miglyol). Ejection fraction, cardiac output, preload recruitable stroke work (PRSW), and end-diastolic pressure volume relationship (EDPVR), are shown; n = number of mice, and *p* values are depicted according to the Student’s *t*-test.

**Figure 3 cells-09-01092-f003:**
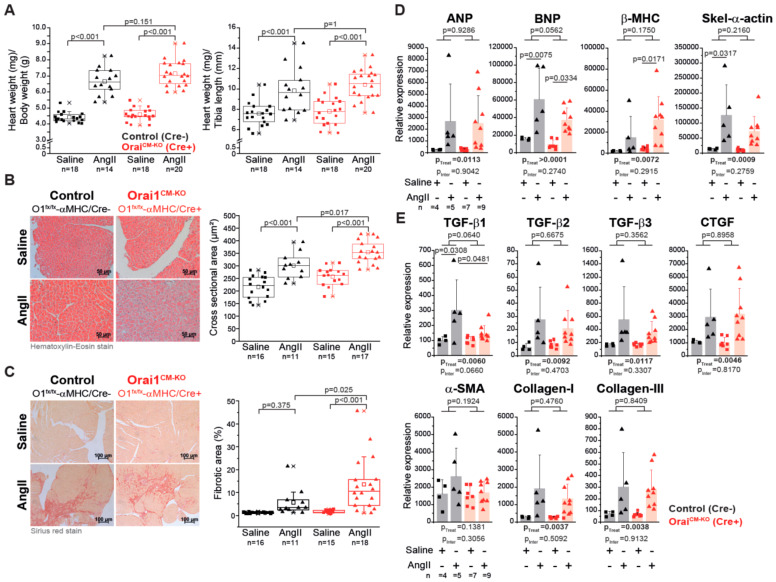
Cardiomyocyte-specific deletion of Orai1 aggravates angiotensin-II-induced cardiac remodeling. (**A**) Analysis of cardiac hypertrophy indexes in Orai1-deficient mice (red, Orai1^flox/flox^/αMHC-Cre^pos^ + tamoxifen) and control mice (black, Orai1^flox/flox^/αMHC-Cre^neg^ + tamoxifen) after angiotensin-II (AngII) treatment. (**B**) The effect of AngII on cardiomyocyte cross-sectional area in Orai1^CM-KO^ mice was analyzed. (**C**) Analysis of the AngII-induced ventricular fibrosis between control and Orai1^CM-KO^ mice is shown. Expression analysis (qPCR) of genes associated with the development of cardiac hypertrophy (**D**) and interstitial fibrosis (**E**) in control (black, Orai1^flox/flox^/αMHC-Cre^neg^ + tamoxifen) and Orai1^CM-KO^ (red, Orai1^flox/flox^/αMHC-Cre^pos^ + tamoxifen) mice after AngII treatment. The n = number of mice, and p-values are depicted according to Bonferroni’s post hoc comparisons after 2-way ANOVA; p_treat_: p-value related to the effect of AngII; p_inter_: p-value of the interaction between AngII and genotype from 2-Way ANOVA analysis.

**Figure 4 cells-09-01092-f004:**
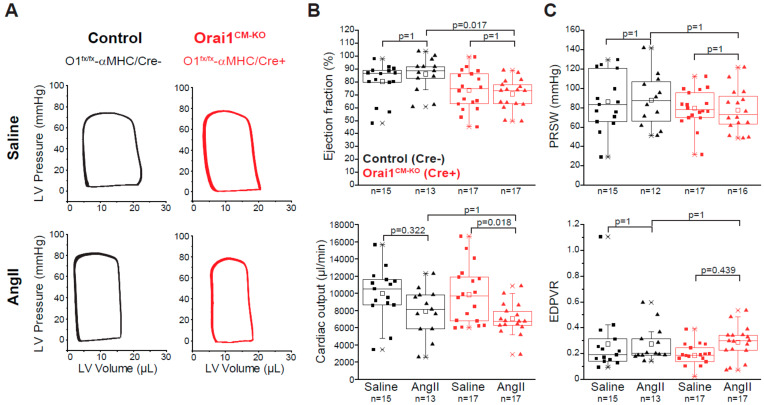
Cardiac function in Orai1 ^CM-KO^ mice after chronic angiotensin-II treatment. (**A**) Representative pressure-volume loops from saline- and AngII-treated control (black, Orai1^flox/flox^/αMHC-Cre^neg^ + tamoxifen) and Orai1^CM-KO^ (red, Orai1^flox/flox^/αMHC-Cre^pos^ + tamoxifen) mice. (**B**–**C**) Analysis of cardiac contractility showing ejection fraction, cardiac output, preload recruitable stroke work (PRSW), and end-diastolic pressure volume relationship (EDPVR). The n = number of mice, and *p* values are depicted according to Bonferroni’s post hoc comparisons after 2-way ANOVA.

**Figure 5 cells-09-01092-f005:**
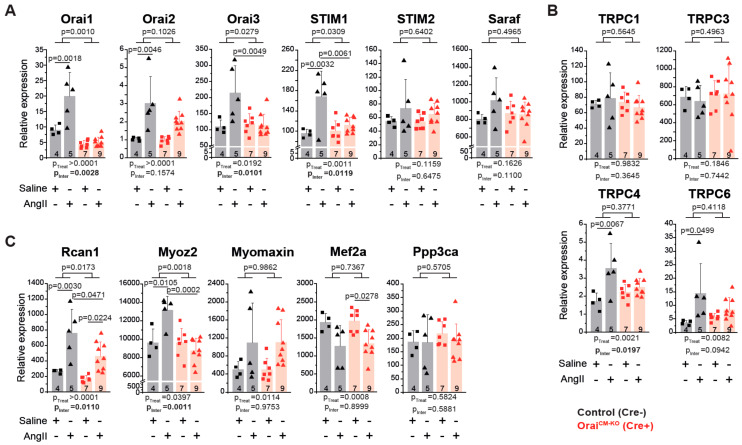
Analysis of Orai1-associated transcriptional regulation in neurohumoral-induced cardiac remodeling. Expression analysis (qPCR) in cardiac samples from control (black, Orai1^flox/flox^/αMHC/Cre^Neg^ + tamoxifen) and Orai1^CM-KO^ (red, Orai1^flox/flox^/αMHC/Cre^Pos^ + tamoxifen) mice via qPCR after AngII treatment. (**A**) Quantitative analysis of Orai1 genes and other genes critical for the store-operated calcium entry. (**B**) Analysis of cation channels from the TRPC subfamily. (**C**) Analysis of genes associated with Ca^2+^-mediated signaling during cardiac remodeling. *Rcan1*: Endogenous calcineurin reporter, *Myoz2* (*Calsarcin-1*): Negative modulator of calcineurin, *Myomaxin*: Transcriptional target of MEF2a, *Mef2a*: DNA-binding transcription factor that activates muscle-specific, growth factor-induced, and stress-induced genes, and *Ppp3ca* (*Calcineurin A alpha*): Cardiac hypertrophy regulator though transcription factors like NFAT. The n = number of mice, and p-values are depicted according to Bonferroni’s post hoc comparisons after 2-way ANOVA.

**Table 1 cells-09-01092-t001:** Pressure-volume analysis in Orai1-WT and Orai1^CM–KO^ mice. Here Orai1^flox/flox^/αMHC/Cre^Pos^ mice treated either with Miglyol (Orai1-WT) or with tamoxifen (Orai1^CM–KO^) were compared.

	Orai1-WT	Orai1^CM-KO^	*p*-Value
	n = 14	n = 10	
Body temperature (°C)	37.19 (±0.56)	37.34 (±0.71)	0.534
HR (bpm)	597.2 (±35.5)	603.1 (±27.0)	0.665
ESP (mmHg)	67.3 (±6.4)	64.4 (±7.7)	0.324
EDP (mmHg)	8.4 (±3.0)	7.4 (±3.3)	0.451
ESV (µL)	8.1 (±5.7)	6.8 (±4.0)	0.530
EDV (µL)	21.8 (±7.8)	22.9 (±5.3)	0.702
SV (µL)	15.1 (±4.8)	17.3 (±2.8)	0.216
CO (µL/min)	9071.6 (±3101.2)	10442.5 (±1798.1)	0.224
Systolic parameters			
EF (%)	71.3 (±16.7)	76.1 (±11.2)	0.442
Stroke Work (mmHg x µL)	883.5 (±317.1)	962.1 (±162.3)	0.482
dP/dt max. (mmHg/s)	6591.9 (±1143.2)	6492.4 (±690.8)	0.809
PRSW	81.4 (±13.2)	89.3 (±16.7)	0.222
ESPVR	17.8 (±18.9)	12.7 (±4.5)	0.534
Diastolic parameters			
Tau (ms)	7.0 (±2.6)	6.3 (±1.1)	0.407
EDPVR	0.16 (±0.07)	0.15 (±0.06)	0.677
dP/dt min. (mmHg/s)	−6679.5 (±1295.3)	−6522.3 (±622.8)	0.727

The n = number of mice, HR: Heart rate, ESP: End-systolic pressure, EDP: End-diastolic pressure, ESV: End-systolic volume, EDV: End-diastolic volume, SV: Stroke volume, CO: Cardiac output, EF: Ejection fraction, dP/dt max.: Maximum pressure increase, PRSW: Preload recruitable stroke work, ESPVR: End-systolic pressure-volume-relation, EDPVR: End-diastolic pressure-volume-relation, dP/dt min.: Minimal pressure increase. Values are given as mean ± SD.

**Table 2 cells-09-01092-t002:** Pressure-volume analysis after angiotensin-II treatment in Orai1-WT and Orai1^CM-KO^ mice. Here Orai1^flox/flox^/αMHC/Cre^Neg^ (Orai1-WT) or Orai1^flox/flox^/αMHC/Cre^Pos^ (Orai1^CM-KO^) mice, both groups treated with tamoxifen, were compared.

	Orai1-WT		p-_1_	Orai1^CM-KO^		p-_2_	p-_3_
	NaCl 0.9%	AngII		NaCl 0.9%	AngII		
	n = 15	n = 13/12		n = 17	n = 17/16		
Body temp.(°C)	36.57 (±0.45)	36.95 (±0.77)	0.803	36.76 (±0.86)	37.12 (±0.52)	0.701	1
HF (bpm)	607.42 (±31.99)	583.5 (±37.78)	0.323	620.58 (±29.65)	614.93 (±29.33)	1	0.060
ESP (mmHg)	63.71 (±9.20)	63.19 (±13.57)	1	64.40 (±10.75)	66.33 (±10.52)	1	1
EDP (mmHg)	6.53 (±3.49)	3.14 (±2.99)	0.052	3.87 (±2.99)	3.11 (±3.61)	1	1
ESV (µL)	4.43 (±2.77)	2.87 (±2.35)	1	6.97 (±4.18)	5.79 (±4.04)	1	0.164
EDV (µL)	19.83 (±5.61)	15.63 (±4.97)	0.319	21.85 (±6.24)	16.11 (±5.42)	0.025	1
SV (µL)	16.46 (±5.16)	13.57 (±4.84)	0.361	16.16 (±5.35)	11.41 (±2.88)	0.021	0.589
CO (µL/min)	9989.75 (±3052.44)	7897.54 (±2726.28)	0.211	10024.73 (±3361.74)	7044.08 (±1856.75)	0.015	0.8417
Systolic parameters							
EF (%)	80.36 (±14.30)	86.02 (±13.32)	1	72.10 (±14.01)	70.69 (±11.55)	1	0.016
Stroke Work (mmHg x µL)	990.61 (±334.67)	863.20 (±447.33)	1	935.46 (±391.33)	669.18 (±250.88)	0.205	0.879
dP/dt max. (mmHg/s)	7738.79 (±885.73)	6650.58 (±1510.60)	0.337	6933.16 (±1810.65)	6231 (±1473.95)	1	1
PRSW	86.26 (±28.95)	87.90 (±27.04)	1	79.32 (±19.44)	77.48 (±21.78)	1	1
ESPVR	14.77 (±8.36)	19.11 (±8.44)	1	15.76 (±11.01)	14.88 (±7.66)	1	1
Diastolic parameters							
Tau (ms)	5.44 (±1.08)	5.42 (±1.24)	1	5.21 (±1.11)	5.66 (±1.80)	1	1
EDPVR	0.27 (±0.25)	0.27 (±0.14)	1	0.19 (±0.09)	0.29 (±0.13)	0.439	1
dP/dt min. (mmHg/s)	−7361.30 (±941.17)	−6643.33 (±1613.47)	1	−6961.90 (±1515.91)	−6545.75 (±1483.59)	1	1

AngII: Angiotensin-II, n = number of mice, Tem.: Temperature, HR: Heart rate, ESP: End-systolic pressure, EDP: End-diastolic pressure, ESV: End-systolic volume, EDV: End-diastolic volume, SV: Stroke volume, CO: Cardiac output, EF: Ejection fraction, dP/dt max.: Maximum pressure increase, PRSW: Preload recruitable stroke work, ESPVR: End-systolic pressure-volume-relation, EDPVR: End-diastolic pressure-volume-relation, dP/dt min.: Minimal pressure increase, p-_1_: p-value of comparison between WT-NaCl and WT-AngII, p-_2_: p-value of comparison between Orai1^CM-KO^-NaCl and Orai1^CM-KO^-AngII, p-_3_: p-value of comparison between WT-AngII and KO^CM^-AngII. Values are given as mean ± SD.

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
