# Peer review of "Cardiomyocyte-Specific Deletion of Orai1 Reveals Its Protective Role in Angiotensin-II-Induced Pathological Cardiac Remodeling"

_cells, 2020, doi:10.3390/cells9051092_

Round 1
Reviewer 1 Report
In the submitted paper, Segin et al. demonstrate that the storage-operated calcium channel Orai1 expressed in cardiac myocytes has some protective role against cardiac remodeling and reduces Angiotensin-II-related myocardial damage. Indeed, cardiomyocyte-specific indicible deletion in adult transgenic mice, while apparently unaffecting baseline cardiac function, leads to a slightly more severe fibrotic and hypertrophic response to angiotensin-II treatment, with some consequences in terms of altered cardiac function.
I have some concerns and requests to the authors.
1)I believe that the experiments performed in embryonic cardiomyocytes are to be deleted from the main article. The effects of angiotensin-II treatment in combination with Orai-1 knockout are completely different when compared with those observed in adult mice. These conflicting results are shown in the first figure and move away the attention of the reader from the main contents of the paper, hampering the understanding of the main message. I strongly suggest to move these results to the supplementary online material and to discuss them only marginally in the m,ain paper, as they do not add much to the results obtained in transgenic mice, and are confirmatory with respect to previous literature.
2)Abstract line 32 "Analysis of cardiac contractility by Pressure-Volume Loops under basal conditions ...did not reveal differences between Orai1CM-KO mice and controls". Introduction line 104: "Analysis of cardiac contractility under basal conditions and after Angiotensin-II infusion showed no differences between Orai1-deficient mice and control mice."
This is the core result of the paper, albeit the experimental evoidence supporting this claim is somehow conflicting. Figure 2D and table 1 show unaltered systolic and diastolic function in response to Orai1 knockout (tamoxifen treatment). However, in the experiment described in figure 4, ejection fraction appears to be reduced by Orai1 deletion, as clearly stated by the authors at line 376: "However, the ejection fraction in Orai1CM-KO mice was reduced independently of the treatment (2-way ANOVA, p= 0.0010)".
This is a severe discrepancy that significantly impairs the significance of the paper, which should be clearly explained by the authors. Are the mice used in the experiments of figure 4/table 2 different from those used in figure 2/Table 1? Why have the authors used different animals? Have the authors used non-inducible Orai-1 KO mice for figure 4/table 2 results?
If the authors used non-inducible Orai-1 mice for Angiotensin-II experiments, I strongly suggest the authors to repeat the experiments with the inducible mice used in Figure2/Table1. Given the role of Orai1 in cardiac development, one cannot exclude complex interactions if the gene is deleted constitutively since birth. in order to demonstrate that Orai1 effectively protects from ang-II remodelling, these experiments must be perfomed on inducible KO mice. In any case, this is unclear from the manuscript and the authors should clearly indicate which mouse line they used for each experiment.
3)The paper lacks any investigation of the mechanisms linking Orai1 KO to cardiac dysfunction after ang-II infusion. In particular, the authors did not perform any fucntional experiments on isolated cardiomyocytes from adult mice after Orai1 deletion with tamoxifen. As Orai1 mediated SR refilling, I qould expect to find reduced SR calcium load and, possibly, altered calcium transients. Why did the authors chose not to perform these experiments?
Reviewer 2 Report
In their article, Segin S., et al. investigated the role of Orai1 protein, which is involved in calcium ions entry to cardiomyocytes, in pathological cardiac remodeling, such as hypertrophy and fibrosis. The authors conducted their studies using either Orai1-deficient mouse strain or conditionally knocked out Orai1 animal model. Thus, the authors could make conclusions about Oria1 involvement in the heart pathology in both neonatal and adult cardiomyocytes. The article is, in my opinion, very well prepared. The results are sound and convincing, analyzed by properly chosen statistical tests. The materials and methods section contains a detailed description of all the experiments, making it easily reproducible. The obtained results are presented in a logical order and depicted properly in five main figures. The discussion is very broad and supported by meaningful references. Overall, I consider this article as very interesting, particularly for scientists seeking for new therapies for cardiovascular diseases.
There are just two little mistakes, which should be corrected:
- Affiliation number 4 is not used. Instead, affiliation 3 is used two times.
- The first word of the discussion is misleading and should be removed (line 449, the word “Authors”).
Reviewer 3 Report
In this original research article, Segin and colleagues investigate the effects of cardiomyocyte-specific deletion of Orai1, a protein required for store-operated calcium entry (SOCE), on cardiomyocyte hypertrophy in vitro and hypertension-induced cardiac pathologies in animal model. For this investigation authors generated cardiomyocyte-specific inducible Orai1 knockout mouse. Authors demonstrate that i) Orai1 deficiency protects embryonic cardiomyocytes from Angiotensin II-induced cellular hypertrophy in vitro; however, Orai1 deficiency in cardiomyocytes is associated with worsen Angiotensin-II-induced cardiomyocyte hypertrophy and fibrosis in mice compared to wildtype mice; ii) Orai1 deficiency in cardiomyocytes is associated with altered expression of genes in Angiotensin II- treated Orai1CM-KO murine hearts that include the genes whose products are involved in SOCE activity, interaction with Orai1 channels, downstream Ca2+ dependent signaling molecules and pathological cardiac matrix remodeling. Based on these observation, authors conclude that cardiomyocyte-specific Orai1 is protective against Angiotensin-II-induced cardiac hypertrophy and fibrosis.
General Comments: The present study is important in the context of pathological cardiac remodeling in response to chronic hypertension and abnormal Ca2+ signaling in cardiomyocytes. However, the study is not entirely novel as the role of Orai1 in cardiac hypertrophy and cardiac remodeling has been studied previously.
Specific Comments:
- In page 2,"Most of the evidence supporting the concept of a role of Orai1 in cardiac remodeling comes from the analysis of cellular models." This statement is not entirely valid because the role of Orai1 in cardiac remodeling has been studied in different animal models.
- In page 6: "In cells isolated from Orai1-/- embryos, the hypertrophic stimulation using Angiotensin-II (AngII) did not result in an increase of cardiomyocyte area, in contrast to Orai1+/+ and Orai1+/- cells, which both showed a significant rise in cell size (Fig. 1B)" There is no change in cell sizes in WT and Hets. In Hets, images are in different magnifications [in Figure 1A Orai1+/- : antiOrai1/anti a-actinin (middle left) and antiOrai1 (middle/mid) are in different magnification]. Please provide both control and AngII-treated images with same magnification in all panels in Fig.1A.
- In page 8: Figure 2, please show the levels of Orai1 protein in tissues by IHC or by Western blot of cardiac tissue lysates. In Figure 2A, the level of Orai3 is significantly decreased. As the levels of Orai3 also controls cardiac hypertrophy and matrix remodeling, please explain the possible contribution of Orai3 in cardiac hypertrophy and cardiac remodeling in cardiomyocyte-specific Orai1 deficient murine hearts.
- While the cross-sectional areas are same in untreated WT and CM specific Orai1 KO hearts in Figure 2C, the cross-sectional areas of CM are increased in untreated CM specific Orai1 KO heart compared to untreated wildtype hearts in Figure 3B. It looks like significantly increased. Please explain the discrepancy of results in Fig. 2C and Fig. 3B.
- In Figure 3E, the level of TGF-β1 is significantly decreased in AngII-treated CM specific KO hearts. The expression levels of other profibrogenic markers α-SMA, COL1 and cytokine TGF-β2 are also leaning towards downregulation. It is difficult to comprehend these results on expression of profibrogenic factors in Orai1 CM KO hearts because under same fibrogenic milieu, the levels of cardiac hypertrophy and fibrosis are further increased (Fig. 3B,C). As profibrogenic cytokine TGF-β is the master regulator of cardiac hypertrophy and cardiac fibrosis; α-SMA is the major collagen producing myofibroblast marker; and type I collagen is the major ECM in the fibrotic tissues, please provide Western blot images showing the levels of TGF-β1, α-SMA, and Type I collagen in ventricular tissue lysates from each group in Figure 3.
- In page 9, "In contrast, histological analysis of cardiac sections revealed that chronic AngII treatment leads to an increase in cardiomyocyte cross sectional area and in fibrosis (2-way ANOVA, p< 0.0001), and that AngII evoked cardiomyocyte hypertrophy (Fig. 3B) and fibrosis (Fig. 3C) were aggravated upon Orai1 deletion in cardiomyocytes." Based on the image presented in Figure 3C, the levels of fibrosis in wildtype and CM specific Orai1 KO hearts are comparable. The Ang II dose used in this study is sufficient to develop significant cardiac fibrosis in WT mice in 2 weeks. However, the quantitative data in Fig. 3C showing AngII-induced stimulation of fibrosis in WT hearts is insignificant compared to controls. Please explain this data. What kind of staining is used in Figure 3B? Is it MT?
- In page 1, "deletion of Orai1 in the adult myocardium reveal a protective function of Orai1 against the development of Angiotensin-II- induced cardiac remodeling, possibly involving signaling via Orai3/STIM1-Calcineurin-NFAT related pathways" As this is simply prediction without any experimental support, authors may provide data showing the alteration of these signaling molecules at the protein levels (total/post-translational modifications) because measuring RNA will not provide any indication on the possible altered signaling in myocardial tissues in the absence of CM-specific Orai1.
- "In our hands, Angiotensin-II alone was effective in inducing hypertrophic effects in cardiomyocytes in WT cells and produced a stronger effect (p<0.001) than phenylephrine after 72h (Fig. S1)." This result could be useful if along with WT cardiomyocytes, Orai1 hets and Orai KO cardiomyocytes are also used.
- In page 13, under Discussion, please delete "Authors"
Round 2
Reviewer 1 Report
The authors addressed all my commets and concerns in a satisfactory manner. I undestand that at the current stage it would not be possible to perform extensive experiments in single adult myocytes. The authors justified the absence of such experiments in great details and I fully accept their justifications. I believe that the changes made by the authors significantly improved the quality of the manuscript.